# Advanced Biomarkers of Hepatotoxicity in Psychiatry: A Narrative Review and Recommendations for New Psychoactive Substances

**DOI:** 10.3390/ijms24119413

**Published:** 2023-05-28

**Authors:** Aniela Golub, Michal Ordak, Tadeusz Nasierowski, Magdalena Bujalska-Zadrozny

**Affiliations:** 1Department of Pharmacotherapy and Pharmaceutical Care, Faculty of Pharmacy, Medical University of Warsaw, Banacha 1 Str., 02-097 Warsaw, Poland; aniela.zabielska1@gmail.com (A.G.); magdalena.bujalska@wum.edu.pl (M.B.-Z.); 2Department of Psychiatry, Faculty of Pharmacy, Medical University of Warsaw, Nowowiejska 27 Str., 00-665 Warsaw, Poland; tadeusz.nasierowski@wum.edu.pl

**Keywords:** hepatotoxicity, biomarkers, psychiatry, new psychoactive substances

## Abstract

One of the factors that increase the effectiveness of the pharmacotherapy used in patients abusing various types of new psychoactive substances (NPSs) is the proper functioning of the liver. However, the articles published to date on NPS hepatotoxicity only address non-specific hepatic parameters. The aim of this manuscript was to review three advanced markers of hepatotoxicity in psychiatry, namely, osteopontin (OPN), high-mobility group box 1 protein (HMGB1) and glutathione dehydrogenase (GDH, GLDH), and, on this basis, to identify recommendations that should be included in future studies in patients abusing NPSs. This will make it possible to determine whether NPSs do indeed have a hepatotoxic effect or whether other factors, such as additional substances taken or hepatitis C virus (HCV) infection, are responsible. NPS abusers are at particular risk of HCV infection, and for this reason, it is all the more important to determine what factors actually show a hepatotoxic effect in them.

## 1. Introduction

Year after year, there is an increasing number of hospitalisations of patients for the abuse of new psychoactive substances (NPSs), commonly referred to as ‘legal highs’. In 2019, in Lancet, some authors indicated that NPSs could be defined as a diverse group of substances that emerged rapidly from the early to mid-2000s. The emergence of newer and newer NPSs in a short space of time and their unknown effect profiles pose a major threat to public health [1]. In 2019–2022, articles were published in an attempt to optimise the pharmacotherapy used in a group of patients abusing mephedrone. One of the main factors increasing the risk of subsequent hospitalisation in patients is liver malfunction. In groups of mephedrone abusers with other psychoactive substances, the highest liver enzyme levels were found in patients with co-occurring HCV infection [2]. Patients participating in a methadone programme, due to mephedrone abuse with heroin, were also re-hospitalised with hepatitis C virus (HCV) co-infection [3]. Among patients with multiple hospital admissions, the number of psychoactive substances taken with mephedrone was greater than one [4]. This is supported by results published in 2020 indicating that supplementation with liver regeneration products may contribute to a reduced risk of subsequent hospitalisation in the same individuals [5]. It should also be borne in mind that due to the abuse of mephedrone with other psychoactive substances, polypharmacotherapy may be one of the factors negatively affecting liver function [6]. For this reason, it seems advisable to carry out research on the relationship between liver function and quality of life in patients abusing various types of NPSs.

The liver parameters studied so far in a larger group of patients, which are simple liver enzymes, are non-specific biomarkers as they are found in many cell types and their increase has been recorded alongside damage to almost every organ. The parameters mentioned include simple liver enzymes such as gamma-glutyltransferase (GGT), alanine aminotransferase (ALT) and aspartate aminotransferase (AST). In order to thoroughly investigate the problem of hepatotoxicity associated with NPS intake, advanced markers of hepatotoxicity, namely, osteopontin (OPN), HMGB1 protein (HMG-1; amphoterin) and glutamate dehydrogenase (GDH, GLDH), should be investigated. This choice of biomarkers was made because of an article published in 2020 in *Frontiers in Pharmacology*, in which the authors describe these biomarkers of hepatotoxicity, the challenges involved and future prospects [7]. OPN levels can predict liver fibrosis and also correlate with the degree of fibrosis, liver failure, portal hypertension and the presence of hepatocellular carcinoma [8]. GLDH is also a useful biomarker of hepatotoxicity, the determination of which can improve the diagnosis of hepatic cell injury [9]. Recent studies have also shown that HGMB1 is a key protein involved in the pathogenesis of acute liver injury and chronic liver disease [10].

This article reviews the indicated biomarkers in psychiatry. The aim of this study was to identify scientific recommendations for future research into the hepatotoxicity associated with the use of new psychoactive substances. Implementation of these recommendations would answer the question of whether new psychoactive substances do indeed have a hepatotoxic effect, or whether co-infection with HCV or taking a series of different psychotropic drugs is responsible for impaired liver function. The results published to date on the concentration of simple liver enzymes do not provide a clear answer. For this reason, this review was written, and recommendations were implemented that should be borne in mind when conducting future research into the effects of new psychoactive substances on liver function.

## 2. Osteopontin (OPN)

Osteopontin (OPN) is one of the main non-collagenous bone matrix proteins [11]. OPN is a glycoprotein that can both initiate and block hydroxyapatite mineralisation in bone tissue [12]. It promotes the proliferation and migration of mesenchymal stem cells, influences osteoblast activity and is involved in the pathogenesis of many bone diseases [13]. In addition to its important roles related to the skeletal system, OPN is important for other systems in the human body, including liver processes. According to Nuñez-Garcia et al., OPN regulates the interplay between phosphatidylcholine and cholesterol metabolism in mouse livers [14]. OPN has specific roles in every cell of the immune system and is also important in acute and chronic inflammation (of the gastrointestinal tract and liver, among others). Iida et al. concluded that OPN alters its association with apoptosis depending on the type of disease and phase of disease activity, acting as a promoter or suppressor of inflammation and inflammatory carcinogenesis [15].

In Figure 1 we outline what we believe to be the two most important mechanisms of OPN. OPN has an affinity for integrins (Figure 1). When OPN binds to integrins, for example, focal adhesion kinase (FAK) is activated, this results in actin remodelling and cell migration. In addition, other processes occur that lead to the activation of phagocytosis. In contrast, the interaction of OPN with the CD44 antigen results in the activation of the NF-κB pathway, which, at the molecular level, initiates the expression of pro-inflammatory genes [16]. Pro-inflammatory genes initiated through activation of the aforementioned pathway include CRP, TNF-α and IL-6 [17].

OPN is a very important cytokine for the liver, as it has both protective functions, e.g., it is involved in liver repair processes, and detrimental ones, e.g., it is an important element in the pathogenesis of numerous liver diseases [18]. According to a review by Song et al., OPN is involved in the processes of liver inflammation and fibrosis and thus plays a major role in the pathogenesis of chronic liver disease [19]. In patients with acute liver failure, plasma OPN levels increase by 10 to 500 times the normal range, and, in addition, these levels correlate with the degree of hepatic necrosis [20]. Furthermore, OPN expression is significantly increased in response to hepatitis, and a change in OPN function may occur due to its cleavage by proteases and thrombin [21]. In the case of liver injury, signalling through the OPN and HMGB1 pathway drives the body’s fibrogenic response [22]. Our work will therefore refer to both of these biomarkers.

OPN is also an important mediator of alcoholic liver disease (ALD), acting through the activation of stellate cells. Inhibition of OPN signalling and OPN receptors partially inhibits the alcohol-induced activation of hepatic stellate cells, plasmin activity and the migration of these cells [23]. A study by Morales-Ibanez et al. found that OPN-deficient mice were protected from alcohol-induced liver damage and showed reduced expression of inflammatory cytokines, such as TNFα, MCP-1 (monocyte chemotactic protein 1) and IL-6 [24]. According to a review by Orman et al., the inhibition of pathways mediated by OPN may be an effective therapy for ALD [25]. According to Banerjee, increased neutrophil infiltration mediated by OPN is a likely contributor to women’s increased susceptibility to alcoholic liver disease [26].

It should be noted that OPN is an important factor in the occurrence and development of cancer. According to a review by Weber et al., it is a marker of tumour aggressiveness and patient survival [27]. Moreover, as reported by Castello et al., OPN appears to be a key determinant of the interaction between tumour cells and the host microenvironment. OPN in the tumour microenvironment is related to the recruitment of leukocyte–endothelial cells and mesenchymal stem cells (MSCs) from the periphery or bone marrow, going further by reprogramming local fibroblasts into cancer-associated fibroblasts. This includes the transformation of anti-tumour M1 macrophages into cancer-associated macrophages [28]. According to Cabiati et al., OPN may be one of the early biomarkers for detecting hepatocellular carcinoma [29]. It is noteworthy that OPN overexpression is associated with intrahepatic metastasis, early recurrence and worse prognosis of surgically resected hepatocellular carcinoma [30]. In addition to the important function of OPN in the skeletal system and in hepatology, the importance of OPN for other systems and diseases within them, including cardiovascular diseases [31] and muscle diseases (e.g., Duchenne muscular dystrophy), has been documented [32]. Undoubtedly, the role of OPN in psychiatry is also an important topic, as we will discuss later in this paper.

### 2.1. OPN in Psychiatry

OPN is also of relevance in psychiatry, wherein it can be considered an advanced biomarker for numerous mental illnesses. Below, we cite reports from the field of psychiatry on the topic of OPN.

Differences in lipid levels have been demonstrated between patients with severe psychiatric disorders and healthy individuals, and links have been discovered between lipid levels and inflammatory mediators, including but not limited to OPN [33]. Higher levels of BDKRB1 (bradykinin receptor B1) and SPP1 (secreted phosphoprotein 1) gene expression were observed in patients after a first psychotic episode (FEP) compared with healthy controls, and those taking risperidone had higher expressions of these genes than all other patients that were studied [34]. Furthermore, schizophrenic patients treated with long-term antipsychotics had significantly lower levels of OPN compared with patients in short-term treatment [35]. A study in which a molecular profiling analysis was carried out in the adult offspring of rat dams given reduced protein in their diets during pregnancy showed an increase in the concentration of OPN in the assayed serum in the test group compared with the control group. These results indicated changes similar to those observed in human schizophrenia [36]. As osteoporosis is observed in schizophrenic patients, OPN levels in schizophrenics were investigated. However, the study found no significant differences between OPN levels in the test and control groups and therefore concluded that mechanisms other than the role of OPN may influence the occurrence of osteoporosis in schizophrenia [37]. Çakici et al. also found no significant differences between OPN levels in schizophrenic patients and healthy controls; however, they found that OPN levels in patients with major depressive disorder (MDD) were significantly lower than in healthy controls [38]. Zhang et al. conducted a preclinical study in which they demonstrated that bone inflammatory markers (including OPN) may play a role in the antidepressant effect of (R)-ketamine, as mice with major depressive disorder (MDD) showed a significant increase in OPN (and OPN/RANKL) in response to (R,S)-ketamine on days 1 and 3 after a single infusion [39]. Moreover, according to a clinical study, patients with MDD have lower serum OPN levels than healthy subjects, and ketamine significantly increased plasma OPN levels on days 1 and 3 after its administration [40]. In contrast, Xiong et al. write about the beneficial effects of (R)-ketamine on a depression-like phenotype, inflammatory bone markers and bone mineral density in a chronic stress model of social failure; however, in this study, OPN concentrations were similar in the test and control groups [41]. It is also worth mentioning a study that showed that in patients with major depressive disorder, electroconvulsive therapy exerts molecular changes in the patients’ serums, including in OPN [42]. Zhang et al. investigated whether the mouse anti-RANKL antibody could attenuate depression-like phenotypes, inflammatory bone markers and bone mineral density (BMD) in mice after chronic social defeat stress (CSDS). Plasma levels of inflammatory bone markers, including osteoprotegerin (OPG), RANKL and OPN, were measured. A single intravenous injection of anti-RANKL (2 mg/kg) induced a rapid antidepressant effect in CSDS-prone mice; however, OPN levels did not change [43]. In another experiment, multiple sclerosis (MS) patients were studied, in which the severity of depressive symptoms (and symptomatic fatigue) was measured using questionnaires, with patients also having their blood OPN levels determined. The results of the study showed that addressing depression (as well as symptomatic fatigue) and increasing physical activity can improve bone mineral density in multiple sclerosis patients. However, OPN concentrations did not differ significantly, as the OPN level (ng/mL) in the study group was 57.0 (15.9), while in the control group, it was 56.3 (12.5) [44]. Proteomic analysis of urine in a mouse model of major depressive disorder (MDD) under chronic unpredictable mild stress (CUMS) showed a decrease in OPN levels in the study group [45]. It is also worth noting a study conducted on cell cultures, which showed that treatment with valproic acid results in an increase in OPN expression after only 4 days of treatment [46]. Ventorp et al. measured the levels of hyaluronic acid and soluble CD44 in the cerebrospinal fluid (sCD44) of suicide attempters (n = 94) and healthy subjects (n = 45). They also investigated other proteins known to interact with CD44, such as OPN and the matrix metalloproteinases (MMP1, MMP3 and MMP9). There were no differences in OPN levels between patients following suicide attempts and healthy subjects [47]. Bone microarchitecture was also assessed in adult women with anorexia, who were found to have higher levels of OPN compared with healthy women, and consequently, women in the study group were more prone to bone fractures [48]. Another study showed that children with autism had significantly higher levels of OPN than healthy children. Elevated serum OPN levels were found in 80.95% of children with autism. OPN levels were significantly correlated with autism severity, such that with increasing autism severity, OPN levels increased [49]. Citing this study, Xu et al. in their paper also identify OPN as a potential biomarker of immunological disorders associated with autism spectrum disorders [50]. Proteomic analysis of urine collected from children with autism also showed elevated levels of OPN compared with healthy children [51]. In HIV-related neurocognitive disorders, cortical neurons have been shown to be a major source of OPN. OPN levels were, furthermore, elevated compared with non-infected individuals and increased with the severity of impairment [52].

Another important aspect is the changes in OPN concentrations in patients who use psychoactive substances, which will be addressed later in this work.

### 2.2. OPN and Use of Psychoactive Substances

Behavioural disorders resulting from the use of psychoactive substances are an important issue, in the context of which research on OPN concentrations has also been carried out, as we will discuss in this section of this paper.

One of the main complications of alcoholism is alcoholic liver disease. This review, which was conducted in 2021, presents the state of knowledge on the contribution of OPN to liver steatosis in the context of alcoholic liver disease and non-alcoholic steatohepatitis. This review reports that serum and liver OPN levels are elevated in patients with alcoholic liver disease. The same results have been provided via studies conducted on animal models. In the course of alcoholic liver disease, increasing levels of OPN inhibit disease progression and iron deposition in the liver. In patients with alcoholic hepatitis, concentrations of full-length OPN are not altered. However, increased levels of OPN fragments are observed, which are broken down by extracellular matrix metalloproteinases [19].

There have been a lot of studies conducted on the topic of OPN related to liver disease resulting from ethanol consumption and dependence. Because of their large number and the fact that the above-mentioned review by Song et al. exists, we have selected and described below the two studies that examined the largest numbers of patients.

Patouraux et al. studied material from heavy drinkers, and the patients were divided into two groups: a retrospective group (109 patients) and a prospective group (95 patients). The authors showed that OPN levels increased in the liver, adipose tissue and serum along with liver fibrosis in patients with alcoholic disease. Furthermore, patients with significant liver fibrosis had significantly higher OPN levels than patients with mild fibrosis [53]. In contrast, Simão et al.’s study included a total of 90 patients, 45 of whom had alcoholic liver disease and 45 of whom had alcoholic cirrhosis and hepatocellular carcinoma. The results showed that OPN is a biomarker of the severity of alcoholic liver disease; however, it is not a marker that could be used in screening for hepatocellular carcinoma [54].

In addition, following the appearance of the 2021 paper mentioned at the beginning, further studies and publications have already appeared on the topic we are discussing, which we will now cite. A preclinical study showed that combining tramadol with chronic alcohol consumption is toxic to the cardiovascular system and is mediated by OPN [55]. Another study showed that intestinal OPN, interacting with the intestinal microflora, has a protective function against alcoholic liver damage [56]. Significantly elevated levels of OPN are observed in serum in the case of alcoholic liver disease, allowing OPN to be considered a diagnostic marker aid [57].

The next most common addiction after the use of psychoactive substances is smoking. Cigarette smoke has been shown to increase OPN expression, thereby contributing to the metastasis of lung cancer [58] as well as the development of pancreatic cancer [59,60,61], emphysema [62,63], pulmonary hypertension [64], bone tissue changes [65,66] and sinusitis in asthmatic patients [67]. Via the increase in OPN expression, it also affects macrophage activation [68] and osteogenic and osteoclastic signalling in the medial palatal suture [69]. In addition, vascular smooth muscle cells that were exposed to nicotine showed increased OPN expression [70].

In heavy smokers with severe asthma, significant differences in OPN levels were observed between the two groups of patients. One of these groups with increased rates associated with Th2 lymphocyte activity had significantly lower OPN levels compared with the group of patients with decreased rates associated with Th2 lymphocyte activity, who had significantly higher OPN levels [71]. According to Maneechotesuwan et al., cigarette smoke extract increases OPN transcription, with simvastatin use inhibiting it [72].

A study by Bishop et al. showed that heavy cigarette smokers who quit smoking for 5 days had significantly lower serum OPN levels compared with those before quitting. A group of 20 patients who smoked 20 cigarettes per day and stopped smoking completely for 5 days were studied, as well as a group of 20 patients who continuously smoked 20 cigarettes per day and did not stop smoking [73]. OPN also plays an important role in the pathogenesis of interstitial pneumonia associated with cigarette smoking, as shown in an experiment by Prasse et al. OPN levels were examined in bronchoalveolar lavage (BAL) cells from 11 patients with Langerhans cell histiocytosis (PLCH), 15 patients with desquamative interstitial pneumonia (DIP), 10 patients with idiopathic pulmonary fibrosis and 5 patients with sarcoidosis, and, additionally, 13 healthy smokers and 19 non-smokers were examined. BAL cells from healthy smokers produced less OPN. Moreover, the BAL cells of healthy non-smoking volunteers did not produce OPN at all. BAL cells subjected to nicotine stimulation increased OPN production [74].

Interestingly, however, a study has emerged in which OPN levels are not significantly different between smokers and non-smokers. The effect of chronic cigarette smoking on the levels of immunological–inflammatory mediators after dental implants in the peri-implant fluid was tested. This is one of the few studies that found no differences in OPN levels between patients who smoke cigarettes habitually and those who do not [75]. We, therefore, believe that this topic requires further research.

Another issue we will discuss in relation to OPN is drug addiction. On this topic, however, the amount of data is severely limited. According to Fu et al., chronic exposure to methamphetamine induces adaptive changes in the brain that underlie the symptoms of addiction, and transmembrane protein 168 (TMEM168) is overexpressed in the caudate nucleus of mice after repeated methamphetamine administration and, in addition, interacts strongly with OPN. The authors of the study suggest that the osteopontin system, regulated by TMEM168, is a novel target pathway for the treatment of methamphetamine dependence, namely, by regulating dopaminergic function in the caudate nucleus [76]. Another study focused on the seminiferous nucleus accumbens taken from mice continuously administered methamphetamine. The results showed that the TMEM168 gene in the seminiferous nucleus interacts with OPN and, moreover, that OPN has a suppressive effect on methamphetamine [77].

We believe that the above examples of cited studies clearly demonstrate the likely importance of OPN in the context of psychoactive substance use and addiction, which calls for further research in this area.

## 3. High-Mobility Group Box 1 Protein (HMGB1; HMG-1; Amphoterin)

HMGB1 protein is released by immune cells and necrotic cells, and secreted HMGB1 activates a number of immune cells, contributing to the excessive release of inflammatory cytokines and promoting processes such as cell migration and adhesion [78]. HMGB1 is, therefore, an important pro-inflammatory cytokine, and it is a chromosomal protein with a structure consisting of three domains, two of which (domains A and B) can assemble autonomously, and there is also a small N domain [79].

In Figure 2 we show that HMGB1 is a ligand for the following membrane receptors, RAGE (receptor for advanced glycation end product), TLR2 and TRL4 (toll-like receptor 2/4), and their activation leads to the generation of inflammatory responses in the body [80]. RAGE—as presented in Figure 2—is a type-I transmembrane protein consisting of three extracellular immunoglobulin-like domains (V, C1 and C2), a single transmembrane helix and a short C-terminal intracellular domain [81] (Figure 2). It is noteworthy that the interaction between HMGB1 and RAGE can induce both pro-inflammatory and anti-inflammatory functions, wherein the functional difference may depend on which companion molecules the RAGE–HMGB1 complex is attached to [82].

Under the influence of the interaction of HMGB1 with TLR2 or TLR4, activation of the full pro-inflammatory response, the NLRP 3 inflammasome [83] and NF-κB and the synthesis of inflammatory cytokines occur [84]. In Figure 2, we also present the structures of TRL2 and TRL4, which are classified as integral type-I transmembrane proteins and are composed of three domains: an N-terminal domain located outside the membrane, a central transmembrane domain (single helix) crossing the membrane and a C-terminal domain located towards the cytoplasm. The extramembrane (N-terminal) domain is terminated in a horseshoe-like shape and has a solenoid structure; it is composed of short leucine-rich tandem repeats and glycan groupings that serve as ligand-binding sites. The structure of the N-terminal domain determines the specificity for a given TRL type [85].

HMGB1 was identified by Wang et al. as a delayed mediator of endotoxin-induced mortality, as the mice tested showed increased serum HMGB1 levels from 8 to 32 h after endotoxin exposure [86]. Additionally, as reported by Scaffidi et al., HMGB1 release can signal cell death to neighbouring cells [87]. HMGB1 acts as an anti-tumour protein, regulating the immune cell response during the process of carcinogenesis, with the implication that abnormal HMGB1 expression is associated with the oncogenesis, development and metastasis of cancer [88]. According to Athavale et al., ablation of HMGB1 in the liver reduces hepatocellular carcinoma but causes hyperbilirubinaemia in mice deficient in Hippo signalling [89]. Other studies indicate that HMGB1 may be an important marker for assessing tumour staging and predicting prognosis in hepatocellular carcinoma, as HMGB1 levels in hepatocellular carcinoma were significantly higher than in chronic hepatitis, cirrhosis and healthy controls. A positive correlation was also found between HMGB1 levels and tumour size [90]. Khambu et al. reviewed the importance of HMGB1 in liver pathogenesis, reporting that the activation of HMGB1 and downstream signalling pathways are contributors to the pathogenesis of non-alcoholic fatty liver disease (NAFLD), alcoholic liver disease (ALD) and drug-induced liver injury (DILI) [91]. A study in mice showed that an increase in hepatic HMGB1 during fibrogenesis contributes to the pathogenesis of the disease by driving the scarring process, so that an increase in HMGB1 is not just the result of liver damage but is also a contributing event and, therefore, a possible target for preventing the onset of liver fibrosis [92]. Jung et al. found that hepatitis C virus infection is inhibited by HMGB1, which is released from virus-infected cells. Secreted HMGB1 may also have a role in cirrhosis, a common comorbidity in patients with HCV [93]. It is worth noting that the HMGB1 protein is important in the pathogenesis of, among others, rheumatic diseases [94], pulmonary diseases [95] and paediatric diseases [96], as well as many others [97]. Undoubtedly, psychiatry is another important field regarding HMGB1 as an important biomarker, as we will discuss below.

### 3.1. HMGB1 in Psychiatry

The role of HMGB1 has been studied in the context of some mental illnesses and has proven to be important for many of them.

Huan Ma reviewed the topic of alarmin levels, including HMGB1, in schizophrenic patients, and he indicates that studies suggest strong links between the HMGB1 protein and schizophrenic disorders. The author of that paper indicates that HMGB1 levels were elevated in schizophrenic patients compared with healthy individuals [98]. Al-Dujaili et al. demonstrated that serum HMGB1 levels were significantly higher in patients with schizophrenia than in a control group of healthy individuals [99]. Very similar findings were also presented by Kozlowska et al. who concluded that HMGB1 is a potential biomarker for schizophrenia and that its levels are also significantly higher in patients with schizophrenia than in healthy subjects [100]. Yilmaz et al. show that serum HMGB1 levels are elevated in schizophrenic patients, irrespective of the phase of the illness, both in the exacerbation and remission phases. Biomarker concentrations were higher in the serum of the study group compared with that of the healthy controls [101]. Furthermore, Mousa et al. confirmed higher levels of HMGB1 in schizophrenic patients and additionally identified HMGB1 as one of the six main predictors of scores on the FF scale (the fibromyalgia and chronic fatigue syndrome rating scale in schizophrenics) [102,103]. Schizophrenia and affective disorders have been found to share a common pathogenesis (related to IL-6, CCL11, HMGB1, DKK1 and KOR), which may explain, at least in part, the symptoms and cognitive impairment observed in these disorders [104]. Other studies have shown that after treating schizophrenics for six months with risperidone, patients’ elevated HMGB1 levels decreased. HMGB1 levels were positively correlated with IL-1β, IL-6, TNF-α and negative disease symptoms [105]. The study by Chen et al. also confirmed elevated HMGB1 levels in patients with a 1st episode of schizophrenia compared with healthy controls, and after 8 weeks of treatment, the study group’s HMGB1 levels decreased significantly; however, they continued to be elevated [106]. In schizophrenia, symptoms of depression and anxiety are driven by immune and inflammatory pathways, wherein HMGB1 plays an important role [107]. According to Al-Hakeim et al., schizophrenic patients responding only partially to treatment, in whom complete remission of the disease cannot be observed following treatment, have increased HMGB1 levels. The lack of complete remission in this group of patients is explained precisely by the elevated levels of HMGB1 [108]. Furthermore, by analysing changes in the expression levels of the HMGB1 gene, which encodes the HMGB1 protein, changes were observed in the expression levels of the HMGB1 gene in the study group (material taken from schizophrenic patients) compared with healthy controls [109]. One experiment that examined the interactions of schizophrenia with 504 different proteins showed that HMGB1 is a protein of importance in the pathogenesis of the disease [110].

In terms of changes in serum HMGB1 concentrations, another fairly well-studied disease entity is depression. The review by Zhang et al. included 69 articles, of which a closer focus was placed on 7 articles that met the criteria set by the authors. HMGB1 levels have been shown to be associated with depression-like behaviours that are similar to motivational deficits [111]. Another review article examines the relationship between depressive symptoms and the expression of inflammatory enhancers. It has been indicated that HMGB1 is one of the inflammatory factors involved in the mechanisms of depression. Administration of anti-inflammatory agents has been shown to alleviate symptoms of depression, indicating the importance of inflammation as a mediator of depression [112]. The review by Liu et al. identifies HMGB1 as a promising therapeutic target for the treatment of depression [113]. In addition, Das et al. propose that the vagus nerve stimulation procedure for treating depression involves inhibiting the production of pro-inflammatory cytokines, including HMGB1 [114]. Ds-HMGB1 (disulfide-HMGB1) and fr-HMGB (fully reduced HMGB) induce depressive behaviour by inducing neuronal inflammation in contrast with nonoxid-HMGB1 (non-oxidizable chemokine-HMGB) [115]. Knockdown or inhibition of RAGE, a receptor to which HMGB1 can bind, may counteract the effects of chronic stress and behaviours that are characteristic of depression [116]. According to Chen et al., acupuncture alleviates depressive behaviour by modulating hippocampal Iba-1 and HMGB1 expression in rats exposed to chronic limiting stress [117]. The prevalence of higher HMGB1 values in people with depression (MDD) has been shown to be higher than in healthy individuals [118,119]. In Hisaoka-Nakashima et al.’s preclinical study, they report that under stress conditions, glucocorticoids induce the release of HMGB1 from astrocytes, leading to a neuroinflammatory state that may mediate major depressive disorder [120]. In addition, it is now known that depression in the offspring of toxoplasmosis-infected mouse mothers can be alleviated with ginsenoside Rh2, which reduces depression by inhibiting microglia activation through an HMGB1-related signalling pathway [121]. Phosphodiesterase-4 (PDE4) inhibition inhibits the HMGB1-related signalling pathway in mice exposed to chronic unpredictable mild stress, and this shows that the downregulation of the HMGB1/RAGE signalling pathway and the suppression of inflammasomes probably contribute to the antidepressant effect of PDE4 inhibitors [122]. Another study in mice indicates that along with depressive behaviour, hippocampal and serum HMGB1 levels increased significantly after 4-week exposure to chronic unpredictable mild stress (CUMS) [123,124]. Sevoflurane was found to exert antidepressant effects by blocking the HMGB1/TLR4 pathway in CUMS-treated rats [125], and curcumin acts in a similar manner [126]. Arctiin also exhibits antidepressant effects by attenuating neuronal inflammation through the activation of NF-κB via the HMGB1/TLR4 and TNF-α/TNFR1 pathways [127]. The same mechanism is used by arctigenin [128], and ketamine exerts its antidepressant effect by acting on the HMGB1/RAGE pathway [129]. Interesting and important information was also provided in the work by Costa-Ferro et al., who found that bone marrow mononuclear cell (BMMC) transplantation prevents depression in rats and modulates inflammatory and neurogenic molecules. HMGB1 gene expression was monitored during the experiment. Overexpression of HMGB1 genes occurred in the hippocampus of animals under chronic mild stress (CMS) but not in healthy controls nor in the stressed but transplanted BMMC group [130]. In contrast, another study showed that, among several genes examined in this context, the gene encoding the HMGB1 protein shows reduced expression and has a significant negative correlation with monocytes [131]. Microglia and neurons appeared to be the main sources of HMGB1-releasing cells in the hippocampus under CUMS (chronic unpredictable mild stress) conditions. According to Wang, minocycline prevents depression-like behaviour by inhibiting HMGB1 release from microglia and neurons [132]. HMGB1-mediated microglia activation induces anxiety- and depression-like behaviours in mice with neuropathic pain [133]. In another experiment, the administration of monosodium glutamate (MSG) induced depression-like behaviour in test animals, with MSG increasing the presence of multiple proteins, including HMGB1. HMGB1 protein levels were elevated in oligodendrocytes in the hippocampus [134]. Quantitative subcellular proteomics of the orbitofrontal cortex of patients with schizophrenia also showed overexpression of HMGB1 (among other things) [135]. The results of a study on lipopolysaccharide (LPS)-induced depression showed that HMGB1 is involved in depression-like behaviours [136]. A subsequent preclinical study showed that HMGB1 infusion into the hippocampus was sufficient to induce anhedonic behaviour [137]. It was also shown that pro-inflammatory cytokines such as HMGB1 were elevated with CUMS, and the use of glycyrrhizic acid (GZA) and quinine (Q) could reduce their levels [138]. As Ghosh et al. write in their paper, HMGB1 inhibition with Gcy (glycyrrhizin) abolishes cognitive dysfunction and disease-like anxiety and depressive behaviour (induced by LPS) [139]. Treatment of mice with GZA, an HMGB1 inhibitor, prevented the activation of enzymes in the kynurenine (KP) pathway and the development of depression-like behaviours [140,141]. Polydatin prevents nervous system inflammation and alleviates depression by regulating Sirt1/HMGB1/NF-κB signalling in mice [142]. Furthermore, ds-HMGB1 has been shown to induce depression in a kynurenine-pathway-related manner and, moreover, the oxidation of fr-HMGB1 induces the activation of the kynurenine pathway, resulting in depressive behaviour [143]. In addition, one study on multimodal psychotherapeutic in-patient therapy for depression found it to be effective in patients with high cytokine production of IFNγ and IL-10, among others, while no significant changes were observed in the levels of other cytokines (including HMGB1) [144].

HMGB1 levels were also significantly higher in patients with bipolar disorder compared with healthy individuals [145,146,147]. In contrast, another experiment suggests that serum HMGB1 levels may be related to the progression of dieting and resistance to food in the course of anorexia treatment [148]. In a mouse model of social failure, icariin and icaritin attenuated neuroinflammation in the hippocampus by mediating HMGB1 expression [149]. It is noteworthy that stress during early life (MS), increases the expression of pro-inflammatory cytokine genes (IL6 and IL1β) and HMGB1 in both the hippocampus and prefrontal cortex of animals, while the use of fluoxetine and exercise during adolescence can decrease the expression of IL6, IL1β and HMGB1 in both the hippocampus and prefrontal cortex in rats exposed to MS [150]. Research has also been carried out on the topic of autism and changes in HMGB1 protein levels, through which some authors are attempting to explain the pathogenesis of autism. Among others, they consider HMGB1 to be an important factor in this process. In a review paper, Di Salvo et al. indicated that alarmins such as interleukin IL-33, HMGB1, heat shock protein (HSP) and S100 protein (S100) may play an important role in the pathogenesis of autism [151]. In recent years, reviews have been published on the role of HMGB1 as an important biomarker in nervous system diseases, as well as in paediatrics, wherein authors have discussed the possible mechanisms by which HMGB1 mediates autism [96,152,153]. Children with autism showed significantly higher serum HMGB1 levels compared with typically developing control children, and these findings suggest that inflammatory processes mediated by HMGB1 may be associated with the specific cognitive characteristics observed in autistic individuals [154]. Interestingly, in children with autism, faecal HMGB1 levels correlated with the severity of gastrointestinal symptoms. The authors propose faecal HMGB1 as a non-invasive biomarker for the detection of gastrointestinal symptoms [155]. A strong correlation was also observed between plasma EGF (epidermal growth factor) levels and HMGB1, suggesting that inflammation is associated with reduced EGF levels [156]. In addition, studies in which adults with autism were the study group show that autistic disorders can be influenced by serum levels of HMGB1 [157,158]. It was discovered that elevated levels of HMGB1 in autistic patients were associated with low levels of zinc and that low zinc levels can cause irritating inflammation in these patients [159]. According to Babinská et al., increased plasma levels of the HMGB1 protein are associated with higher rates of gastrointestinal dysfunction in individuals with autism [160]. In contrast, according to a review by Dipasquale et al. [161], elevated levels of HMGB1 are observed in the blood of young and adult patients with autism spectrum disorders. The literature review conducted in the article shows that only a few studies have assessed serum HMGB1 levels in people with ASD and that all of these have been conducted on small samples [161]. Elevated plasma epidermal growth factor receptor (EGFR) levels were also observed in autistic children, and this correlated with symptom severity in autistic children, as well as with HMGB1 levels [162].

Since exposure to air pollution during pregnancy increases the risk of autism spectrum disorders, a study was conducted on mice that were exposed throughout pregnancy to nano-sized particulate matter (nPM). Gestational exposure to nPM has been shown to alter HMGB1 signalling from early development into adulthood [163]. According to Tianliang Zhang, gestational exposure to PM2.5 leads to cognitive dysfunction in mouse offspring by promoting hippocampal inflammation via the HMGB1–NLRP axis [164]. In addition, one theory is that infections with certain viruses can lead to behavioural disorders, the mechanisms of which are linked to HMGB1 functions [165]. As can be seen from the examples above, HMGB1 may be an important biomarker of psychiatric diseases. We believe that it may also be relevant in the context of psychoactive substance use, as we will discuss later in this paper.

### 3.2. HMGB1 and Use of Psychoactive Substances

In this section of our work, we will cite relevant studies that address the role of HMGB1 during psychoactive substance use.

Crews et al., writing a paper on mechanisms that rely on the induction of neuroimmune genes in alcoholics, focus a lot on the release of HMGB1 and the expression of genes encoding HMGB1. The authors cite as one of their main conclusions that chronic alcohol consumption contributes to an increase in HMGB1 expression, resulting in neurodegeneration and changes in the brains of alcoholics, such as disruption in the neuronal network and death of some neurons. The expression level of HMGB1 is correlated with lifetime alcohol consumption and the age at which patients started drinking alcohol. Crews et al. cite available publications on HMGB1 and alcoholism [166].

Another important aspect related to HMGB1 and compulsive alcohol consumption is alcoholism-related liver disease. In 2018, Gaskell et al. reviewed HMGB1 and liver disease, with much focus on alcoholic liver disease. HMGB1 acts as a pro-inflammatory cytokine that contributes to liver damage in a variety of liver diseases, including alcoholic liver disease. After alcohol consumption, oxidative stress occurs in the body, resulting in hepatocytes actively secreting HMGB1 in the damaged liver. Various isoforms of HMGB1, such as disulphide-linked hyperacetylated HMGB1, were observed in the serum of both ethanol-treated mice and patients with alcoholic liver disease. Hepatocytes have been shown to be the source of these isoforms; thus, HMGB1 derived from hepatocytes is thought to be involved in the pathogenesis of alcoholic liver disease [10].

As in the chapter on OPN and alcohol dependence, due to the large number of publications, as well as the two articles reviewing this topic described above, we select and describe below the two studies that surveyed the largest number of people from the available literature.

Orio et al. showed that there are gender-related differences in the peripheral inflammatory response to alcohol in young people, with increased levels of alcohol-risk-related molecules such as HMGB1 observed in female drinkers. Furthermore, higher levels of inflammatory markers, including HMGB1, correlated with poorer performance in episodic memory and executive functioning tasks in female drinkers compared with male alcohol drinkers. A total of 42 people aged around 20 years took part in the study (of whom 22 non-drinkers constituted the control group and 20 drinkers the study group). Immunological/inflammatory changes were more prominent in female drinkers. In this group, levels of inflammatory markers correlated with poorer performance in episodic memory and executive function [167]. The second paper we choose to discuss is the study by Vannier et al. Studying 80 patients with alcoholic liver disease, the authors demonstrated that serum HMGB1 levels were positively associated with the severity of liver disease. In addition, HMGB1 levels in these patients effectively predicted hospital readmission for liver disease as well as transplantation or death within 90 days [168].

Following the publication of the above-mentioned reviews, more recent studies on HMGB1 and excessive alcohol consumption have already appeared, which we will now discuss. The HMGB1/TLR4 signalling pathway is being considered as a potential and promising therapeutic target, particularly for patients with alcoholic liver disease [10,169].

HMGB1 is also an important biomarker in the context of smoking addiction. Several review papers have been produced on this topic, which we will now refer to. Almost all the studies referred to in the 2015 review by Gangemi et al. showed that HMGB1 levels are elevated in smokers and COPD (chronic obstructive pulmonary disease) patients. Compulsive cigarette smoking is the most common cause of COPD, which, through neutrophil death, leads to the release of HMGB1 [170]. Another review reports that the NLRP3, NLRP6, NLRP12 and AIM2 inflammasomes are important elements in the pathogenesis of several smoking-related diseases and that HMGB1 molecules are involved in the activation of these inflammasomes [171].

The 2022 reviews are concerned with the receptors for which HMGB1 has an affinity (RAGE and TLR4). The first relates to COPD, which is triggered by cigarette smoking, and the role of the HMGB1/RAGE/TLR4 signalling pathway in this process. Cigarette smoke can cause lung inflammation, and HMGB1 and its receptors are involved in the activation of airway inflammation, which in chronic exposure is very likely to lead to the development of COPD [172]. The second also deals with COPD, for which cigarette smoke exposure is a factor in its development, but focuses only on the role of the RAGE receptor and its ligand HMGB1. The authors indicate that the RAGE receptor is a determinant of susceptibility to COPD as well as a genetic marker of low lung function [173].

Due to the existing reviews indicated above, on the topic of smoking addiction and HMGB1, we also selected the two studies that involved the largest numbers of people. The first showed that HMGB1 translocation and release, induced by cigarette smoke, contributes to NF-κB migration and activation through the induction of autophagy in lung macrophages. In that study, the researched group comprised 15 cigarette smokers and 15 COPD patients, and the control group comprised 15 healthy non-smokers [174]. The second study looked for an association between HMGB1 and EGFR mutations in a group including 280 patients with non-small cell lung cancer (NSCLC), some of whom were smokers and others who had never smoked. The results suggest that HMGB1 polymorphisms are significantly inversely associated with EGFR mutations among smoking patients with NSCLC and, furthermore, that HMGB1 variants and smoking may contribute to the pathological development of NSCLC [175]. Following the publication of the above-mentioned reviews, another study was published, which claims that chronic cigarette smoking may contribute to cognitive impairment arising via the HMGB1-RAGE/TLR4-NF-κB pathway. According to the authors of the study, GLP-1-mediated cortactin has a neuroprotective effect by inhibiting the aforementioned pathway [176]. Moreover, elevated serum HMGB1 levels have been observed in chronic e-cigarette users [177]. HMGB1 is also involved in the pathogenesis of bronchial asthma induced by tobacco smoke in animal models [178].

Studies on drug uptake also show that HMGB1 may have a role in the development of psychoactive substance dependence. According to Gao et al., neuronal HMGB1 in the nucleus accumbens regulates reward memory after cocaine [179]. In contrast, Frank et al. showed that the hazard-related molecular pattern HMGB1 mediates the neuroinflammatory effects of methamphetamine use [180]. Other studies also show that methamphetamine injection induces hyperthermia, an increase in plasma HMGB1 concentration, the degeneration of dopaminergic nerve endings, microglia accumulation and the extracellular release of neuronal HMGB1 in the striatum [181]. The results of a meta-analysis that showed that methamphetamine exposure increases beta-amyloid precursor protein expression through an HMGB1-related pathway in Alzheimer’s disease patients appear to be interesting [182]. HMGB1, which is involved in methamphetamine-induced astrocyte activation and migration, may be a good therapeutic target to extinguish these changes in astrocytes [183].

We believe that the above examples clearly demonstrate that HMGB1 has an essential role in psychoactive substance use disorders.

## 4. Glutamate Dehydrogenase (GDH, GLDH)

In a review on the role of GDH in cell and tissue biology, Plaitakis et al. point out that GDH is a hexameric enzyme that catalyses the reversible conversion of glutamate to α-ketoglutarate and ammonia while reducing NAD(P) + to NAD(P)H [184]. We have illustrated this process in Figure 3, wherein the amino group (highlighted in red) is detached, and glutamate is deaminated to α-ketoglutarate [185] (Figure 3).

Hepatic GDH controls whole-body energy partitioning through amino-acid-derived gluconeogenesis and ammonia homeostasis [186]. Thus, GDH is important for ammonia binding and amino acid homeostasis in the brain during hyperammonaemia [187]. As early as in 1964, Frieden et al. wrote that GDH activity is strongly and specifically influenced by purine nucleotides, particularly adenosine and guanosine di- and triphosphate, but interestingly, GDH from non-animal sources (microorganisms or plant leaves) is not influenced by purine nucleotides [188]. According to Desai et al., low levels of GDH are associated with clostridium difficile colonisation in patients with inflammatory bowel disease who have tested positive for PCR [189]. A study on the prevalence of clostridium difficile infection (CDI) among hospitalised patients with inflammatory bowel disease in Greece found that of 6932 patients screened for CDI, 894 patients were positive for GDH (12.89%) [190]. In addition, GDH is known to be a biomarker of mitotoxicity following acute liver injury, and additionally, the ratio of GDH to ALT (alanine aminotransferase) may provide insight into the role of mitochondrial damage in liver injury [191]. Other studies have shown that fluoride exposure induces damage to mitochondria and the liver ultrastructure and increases GDH levels [192]. Restricted nutrition modulates diurnal changes in the activity, expression and histological localisation of GDH in the liver [193]. In contrast, in a review that looked at diagnostic markers in liver damage from drugs, herbs and alcohol, GDH was identified as a potential idiosyncratic biomarker of drug-induced liver injury (DILI) [194]. GDH is presented in numerous papers as a promising biomarker of DILI that occurs after pharmacotherapy with paracetamol [195], isoniazid [196] or furosemide, among others [197]. It is also worth noting the important function of GDH in oncology. In a study by Michalak et al., lung cancer chemotherapy induced an increase in glutamate in peripheral blood mononuclear cells (PBMCs), and its serum concentration increased GDH activity in PBMCs. The results show that the initial symptoms of neurological deficits, as well as new symptoms that are a complication of chemotherapy, are related to changes in markers of glutamate metabolism [198]. According to Spinella, metabolic recycling of ammonia by GDH supports breast cancer biomass, as in the mice studied, ammonia accumulated in the tumour microenvironment and was directly used to generate amino acids through GDH activity [199]. Furthermore, there are reports that the mitochondrial enzyme GDH1 is commonly elevated in human cancers. In [200], the authors point to lung cancer and breast cancer.

Psychiatry is another field in which GDH activity can undoubtedly play an important role, as we will discuss later in our paper.

### 4.1. GDH in Psychiatry

Levels of GDH activity have been studied in patients with certain mental illnesses and were found to be significantly altered in psychiatric patients compared with controls, as we will discuss in this chapter.

The largest number of studies on the importance of GDH in psychiatry relate to schizophrenia. Savushkina et al. found that baseline GDH levels could serve as a predictor of the efficacy of antipsychotic therapy in patients with schizophrenia, as in that study, GDH activity significantly increased during post-antipsychotic treatment [201]. According to Burbaev et al., treatment with olanzapine affects the amounts of glutamate-metabolising enzymes in the platelets of patients with chronic schizophrenia. It was also found that the longer the disease lasted before starting the treatment cycle, the higher the GDH levels measured after treatment [202,203]. Baseline platelet GDH activity may serve as a predictor of the efficacy of antipsychotic therapy in patients with schizophrenia [201]. A review by Plaitakis on the topic of GDH deregulation in neurological diseases showed that GDH deregulation in schizophrenia remains an important issue to be addressed [204]. Lander et al. indicated that impaired GDH function is important for psychiatric and neurological disorders [205]. Both glutamate uptake by astrocytes and its mitochondrial catabolism initiated by glutamate dehydrogenase are important nodes of glutamatergic regulation in astrocytes, and alterations in these processes may contribute to metabolic deregulation in schizophrenia [206]. A study on the three GDH isoforms performed on brain tissue homogenates from deceased individuals with schizophrenia found increased GDH I + II activity compared with healthy controls (in the caudate nucleus and cerebellum), while GDH III activity was not significantly different between the groups [207]. In contrast, another study observed altered GDH levels in the prefrontal cortexes of people who had schizophrenia compared with controls. Immunoreactivity of the three GDH isoforms was significantly elevated in patients who had schizophrenia. According to the authors, the increased levels of GDH enzymatic activity in the aforementioned area in the study group are due to an increase in all three levels of the GDH isoenzymes [208]. GDH enzymatic activity was also significantly increased in the prefrontal cortex, posterior cingulate cortex and cerebellar cortex in patients with schizophrenia compared with controls. According to Tereshkin et al., the alterations in the levels of immunoreactive forms of GDH in the brains of patients with schizophrenia is one of the causes of impaired glutamate metabolism in the brain and an important aspect of the pathogenesis of schizophrenia [209]. Altered levels of GDH, among other things, in the brains of patients with schizophrenia is one of the causes of impaired glutamate metabolism in these brain structures and an important aspect of the pathogenesis of schizophrenia [210]. According to Burbaev et al., an increase in GDH levels was observed in certain parts of the cortex in patients with schizophrenia. The authors believe that an increase in GDH-I immunoreactivity in schizophrenics compared with control subjects was observed in the frontal cortex and posterior cingulate cortex [211]. The same authors demonstrate that changes in the expression of glutamine synthetase, a glutamine synthetase-like protein and three GDH isoenzymes in the frontal cortexes of patients with schizophrenia suggest that there is impaired glutamate metabolism in this mental illness and Alzheimer’s disease [212,213]. In a preclinical study, it was discovered that after ketamine administration, GDH activity was reduced in all rat brain areas tested. Clozapine treatment resulted in a significant increase in GDH activity levels in all three brain areas compared with the effect of ketamine [214]. GDH-deficient mice show behavioural abnormalities similar to those in schizophrenia and hippocampal dysfunction in the CA1 subregion, the overactivity of which is specific to schizophrenic patients. GDH is encoded by Glud1, and Glud1-deficient mice have behavioural abnormalities that are characteristic of schizophrenia. A study found a significant post-mortem reduction in Glud1 expression in the CA1 subregion associated with schizophrenia [215]. However, when examining the implications for altered glutamate and GABA metabolism in the dorsolateral prefrontal cortexes of older patients with schizophrenia, no significant differences were observed in the levels of GDH activity in these areas between the test and control groups [216]. In late-onset schizophrenia and delusional disorders, changes in glutamate and glutathione metabolism, including changes in metabolic enzymes such as GDH, affect the course of the illness [217]. It was also shown that serum GDH activity tended to be lower in people with schizophrenia with late dyskinesia than in people with schizophrenia without it [218]. In a preclinical study, GDH mRNA was found to be immediately induced after phencyclidine treatment in rat brains. The results of the experiment suggest that GDH mRNA induction may be involved in the pathology of PCP-induced psychosis (phencyclidine-induced schizophrenic psychosis) and that the gene encoding GDH may be one of the candidate genes that are susceptible to schizophrenia [219]. Prokhorova et al. showed significant differences in pre-treatment GDH activity between subgroups of patients with a first episode of psychosis (FEP), chronic patients and controls, through which they concluded that baseline levels of platelet GDH activity may be important in predicting the efficacy of antipsychotic pharmacotherapy in patients with FEP [220]. In male patients with depression, it was found that baseline GDH activity levels were not only reduced in patients compared with activity levels in healthy controls but also differed within the study group according to the type of prevailing symptoms, either positive or negative [221]. In addition, in another experiment, depressed patients showed a significant decrease in platelet GDH activity compared with healthy controls [222]. Glutamate metabolism enzyme activity, including GDH in first-episode juvenile depression with attenuated schizophrenia symptoms, has also been shown to be a significant biomarker in the treatment of the illness [223]. In a study on hospitalised patients with mild–severe depressive episodes, blood samples were taken from the patients to determine GDH activity. The study involved three groups of subjects (Cl.1, Cl.2 and Cl.3) that differed in biochemical indices, with Cl.2 patients (n = 11, 20.8%) having increased COX activity and decreased GDH activity. It has been shown that there is a relationship between the nature of changes in metabolic parameters and differences in the course of late-onset depression [224]. As reported by El-Ansary et al., platelet GDH activity was reduced in their study group, i.e., patients with autism, compared with controls. The study involved 20 boys with autism aged between 4 and 12 years and 20 healthy controls. A decrease in GDH activity levels was observed in the study group, but this was not a statistically significant result [225]. A statistically non-significant reduction in GDH activity levels in autistic individuals was also observed by Shmais et al. [226]. Furthermore, another study that assessed (post-mortem) GDH expression levels in the brains of autistic and healthy subjects showed no statistically significant differences between the groups [227]. The phenomenon of increased liver enzymes (including GDH) with four different neuroleptics, haloperidol, clozapine, perphenazine and perazine, was also investigated. The results showed the incidence of each abnormality in the enzyme values (exceeding three-fold or two-fold the normal range) after the administration of each drug; the most common abnormality was observed after haloperidol administration [228]. Another study showed that the inhibition of the enzymatic activity via psychotropic drugs of the GDH isoenzymes (GDH1 and GDH2) allows haloperidol to be used to regulate GDH activity and glutamate concentration in the central nervous system [229]. Chronic use of haloperidol may contribute to increased GDH activity [230]. We think it is also worth noting that the Glud1 gene is associated with susceptibility to schizophrenia, autism, depression and bipolar affective disorder. GluD1 gene knockout mice show impaired fear memory and social interactions and enhanced depression-like behaviour [231]. In our work, we only mention an important aspect of this gene; however, this is a very extended topic that should be covered in a separate publication.

These examples show that GDH can be recognised as an important biomarker of mental illness.

### 4.2. GDH and Use of Psychoactive Substances

GDH is also a subject of research and an important biomarker in the context of psychoactive substance use, as well as substance abuse, which will be discussed in this section of this paper.

On the topic of alcoholism, as with the biomarkers discussed above (OPN and HMGB1), we also chose two studies with the largest study groups. In these studies, which we will report on below, the authors reviewed the knowledge of GDH and compulsive alcohol consumption in the theoretical sense. Kravos et al. [232] point out that the available studies mentioning GDH as a potential biomarker have been contradictory, despite their large number, examples of which include the work by Chemnitz et al. [233], Hussain et al. [234] and Conigrave et al. [235]. The publications by Kravos et al. have dispelled many doubts on this topic.

According to Kravos et al. [232], GDH is a sensitive biomarker of alcoholism, with levels falling almost immediately after the cessation of drinking, which is much faster than is the case with other known biomarkers such as gamma-glutamyltranspeptidase (GGT), aspartate aminotransferase (AST), alanine aminotransferase (ALT) and mean erythrocyte volume (MCV). The levels of these biomarkers were assessed 3 times in 238 patients (alcoholics) immediately after hospital admission and 24 h and 7 days after hospital admission [232,236].

The second study we wish to refer to, which also has the largest number of people studied of the data available on the subject, is the work also cited by the previously mentioned Kravos et al. In 1980, Worner et al., investigating the time course of GDH activity in the plasma of alcoholics, showed that GDH values tended to be highest during or immediately after cessation of drinking, followed by a rapid decline towards normal values, and, importantly, early GDH values correlated well with liver histology in patients who subsequently underwent diagnostic liver biopsy. The study group consisted of 42 alcoholics admitted to hospital for detoxification and, in addition, 2 volunteers consuming ethanol in a specified amount in hospital were studied [237].

On the topic of cigarette smoking, we found no articles that examined GDH concentrations or activity in smoking-dependent individuals. The only study we found that addresses this area focuses on the effect of maternal smoking on foetal liver protein expression during the second trimester of pregnancy (in a sex-dependent manner) and showed that maternal smoking affected foetal homeostasis, including levels of type-1 GDH activity. Liver extracts from 55 human foetuses in the 2nd trimester (week 12–17) of pregnancies that were electively aborted were studied [238]. In the context of drug use, we will recount (as in the case of OPN and HMGB1) all the available data in the literature. GDH plays an important role in metabolism, the overdose mechanism and drug dependence. This is because the alkaloids in tobacco, cocaine, medical poppy and cannabis smoke are potent nucleophiles targeted by the Schiff base, which is an intermediate complex between α-ketoglutarate and GDH [239]. According to Shi et al., GDH type-1 (but not GDH type-2) polymorphisms are associated with heroin addiction. It has been shown that in the treatment of heroin addiction, the dose of methadone used in therapy depends on which GDH genotype the patient presents [240]. In contrast, Vicente-Rodríguez et al. studied the phosphoproteins in the striatum of pleotrophin knockout mice and midkine knockout mice treated with cocaine. Among the seven phosphoproteins identified as being altered in the study group was GDH. The experiment showed that GDH underlies cocaine-induced neurotoxicity and neurodegeneration. For this reason, in the future, GDH may be one of the new therapeutic targets for cocaine addicts, which currently still requires a lot of research [241]. According to a study by Fasakin et al. [242], alkaloid extracts from hemp (Cannabis Sativa) and noble tobacco (Nicotiana tabacum) have anti-anxiety and antidepressant effects, and their mechanism of action involves increases in GDH activity, dopamine and IL-10 levels. The authors write about the therapeutic potential of these extracts but also point out the obvious potential for patients to become dependent on them [242]. In addition, Ghoneim et al. in one of their studies assessed GDH activity, which appeared to be peripherally increased in rabbits that were fed hashish for one month [243].

We believe that the above examples indicate that GDH may also be important in patients using or addicted to psychoactive substances.

A list of the most important research on GDH, OPN and HMGB1 in mental illness is shown in Table 1.

## 5. Recommendations for Hepatotoxicity Testing of New Psychoactive Substances

The first recommendation concerns other substances taken with NPSs. The studies conducted to date on the effects of taking different types of NPSs are mainly based on single preclinical studies or on the examination of non-specific parameters such as AST, ALT and GGT. For this reason, it seems advisable to investigate the advanced markers of hepatotoxicity described in this review in patients abusing NPSs with other drugs or alcohol. In a study by Kravos et al., it was shown that GDH is a sensitive biomarker of alcoholism, the level of which decreases following the cessation of alcohol consumption [232]. Worner et al. showed that GDH levels showed an association with liver histology in patients who subsequently underwent diagnostic liver biopsy [237]. A 2021 review of OPN found that serum and liver levels of this biomarker are elevated in patients with alcoholic liver disease [19]. The same is true for HMGB1 expression levels correlated with alcohol consumption [166]. One of the main substances taken with NPSs, such as mephedrone, is alcohol [244]. Investigating the GDH levels in patients abusing NPSs with alcohol in relation to alcohol-only patients would help answer the question of whether it is the new psychoactive substance that has a hepatotoxic effect or whether it is, conversely, the alcohol consumed that has a greater negative impact on liver function. One of the most commonly taken substances along with NPSs is heroin. Addiction to this psychoactive substance is strongly associated with GDH type-1 polymorphisms. The dose of mephedrone used in the treatment of heroin addiction shows a relationship with the patient’s respective GDH genotype [213]. In the studies to date on patients participating in a methadone programme due to heroin mephedrone abuse, it has been shown that one of the main factors increasing the risk of subsequent hospitalisation is HCV infection [3]. However, only simple liver enzymes were studied in that article. For this reason, GDH levels would need to be investigated in addition to relevant reference groups, including patients with HCV infection only. Among other things, this would make it possible to check whether or not liver function deteriorates with an increase in the dose of methadone taken.

The next recommendation is to investigate the effect of psychotropic drug use on the levels of advanced markers of hepatotoxicity in patients abusing new psychoactive substances. According to data published in *Pharmacological Research*, one of the main factors increasing the risk of subsequent hospitalisation for patients abusing NPSs with other substances may be the polypharmacotherapy used. The use of a range of psychotropic drugs increases the risk of associated side effects, which may consequently hinder the therapeutic effect [245]. For this reason, it is advisable to test for advanced markers of hepatotoxicity before starting treatment in a patient taking a particular NPS and then several times during the course of the treatment being administered, while being mindful of whether the drug being taken may adversely affect liver function. This would make it possible to determine whether taking psychotropic drugs has an additional hepatotoxic effect or whether, on the other hand, worsened liver function was present during the patient’s admission to hospital, i.e., before the implementation of treatment. Psychotropic drugs with potential hepatotoxic effects should not be used in patients with suspected liver abnormalities [246], and such patients may be NPS abusers. The same is true for investigating the effect of liver regeneratives taken that may reduce the risk of subsequent hospitalisation of the same patients [5].

Another recommendation is to test the levels of advanced markers of hepatotoxicity in patients abusing NPSs with co-occurring depressive behaviour. Psychoactive substance use may be related to the response to different types of stressful situations [247]. In published case reports of patients abusing NPSs, depressive disorders are often described [6]. In a review on HMGB1 levels, an association with depression-like behaviours that are similar to motivational deficits was found [111]. This biomarker appears to be a promising therapeutic target for the treatment of depression [96]. For this reason, one additional recommendation is to measure HMGB1 in patients taking NPSs, as its inappropriate levels may be responsible for an increased risk of depressive disorders. The group of patients abusing various types of NPSs is mainly male. One of the studies mentioned in this review indicates that a reduction in GDH levels was observed in men suffering from depression compared with a control group [198]. It is also worth noting that impaired fear memory and social interactions and the reinforcement of depression-like behaviours were observed in GluD1 gene knockout mice [208]. A study on 3023 marginalised, nocturnal and online NPS users found that they reported, among other things, increased medium- and long-term mental and physical problems and more social problems [248]. A similar recommendation applies to schizophrenia. An example herein is the case report of a patient who took NPSs such as ‘el blanco’ while on leave, which consequently contributed to the onset of schizophrenia relapses [249]. Schizophrenic patients have elevated levels of HMGB1 compared with controls. This is related to the strong association of the HMGB1 protein with schizophrenic behaviour [80]. Another argument is that in schizophrenic patients, baseline GDH levels may serve as a predictor of the efficacy of antipsychotic therapy [177]. Long-term use of antipsychotics compared with short-term treatment has a significant reductive effect on OPN levels [36]. For this reason, investigating advanced markers of hepatotoxicity in patients abusing NPSs could answer the question of whether their levels show statistically significant associations with the severity of psychiatric disorders.

Summarising the recommendations indicated above, it is recommended to investigate the indicated advanced markers of hepatotoxicity in patients abusing various types of NPSs with other substances in the future for several reasons. Firstly, with the inclusion of appropriate comparison groups, it would be possible to answer the question of whether it is indeed NPSs that exhibit hepatotoxic effects or whether it is the additionally ingested substances that do so. Secondly, due to the high prevalence of HCV infection in the patient group in question, a group of people with this type of infection only, i.e., not taking any psychotropic substances, should also be included in a future study. Thirdly, investigating the effects of the psychotropic drugs used in patients abusing NPSs would allow us to check whether they have additional hepatotoxic effects and are therefore increasing the risk of subsequent hospitalisations for the same patients. One more argument pointing to the necessity of testing OPN, GDH and HMGB1 levels is to relate the results obtained to the patient’s mental state measured with appropriate scales and questionnaires, thus increasing the chance of obtaining a therapeutic effect, such as patients stopping taking NPSs.

The limitations of the studies carried out so far lie primarily in the investigation of simple liver enzymes in patients abusing various types of NPSs. For this reason, it is difficult to determine the effect of their intake on liver function. It is essential that future studies take into account factors such as the pharmacotherapy used, additional psychoactive substances taken with NPSs, and HCV infection. Only then will it be possible to determine whether the various types of NPSs have a hepatotoxic effect or whether another factor plays a major role. Multiple hospitalisations due to NPS abuse pose a major public health challenge, and one important predictor may be inadequate liver function. For example, one study found that predictors of thirty-day re-hospitalisation in patients with decompensated cirrhosis included elevated liver enzymes and hepatic encephalopathy [250]. Another example includes taking opioids, which can contribute to chronic liver disease (CLD) and increase the risk of subsequent hospitalisations [251]. A 2019 article published in *Frontiers in Psychiatry* pointed out, among other things, that the chances of hospital readmission increase when multiple substances are taken concurrently, which is characteristic of NPS use [252]. In a 2017 article published in *BMC Psychiatry*, the authors found that in a group of people with substance use disorders, hepatitis C was associated with an increased risk of hospital readmission [253]. The use of polypharmacotherapy alone may cause subsequent hospitalisations because of hepatotoxic effects often resulting from drug interactions. A study by Kadra et al. found that antipsychotic polypharmacy increased the risk of hospital readmission within six months compared with patients who received monotherapy [254]. The concomitant intake of a number of NPSs with other substances showing hepatotoxic effects, combined with the adverse effects of multiple psychotropic drugs, can only further increase the risk of subsequent hospital admissions. This is all the more reason why the advanced markers of hepatotoxicity discussed in this manuscript should be investigated in order to study the real impact of taking NPSs on liver function. Measurements should be taken several times starting from the moment a patient is admitted to hospital, during hospitalisation and ending with his discharge home. It is also important to bear in mind the necessity of supplementing patients abusing NPSs with liver regeneration preparations, as the proper functioning of this organ may subsequently contribute to improving the metabolism of the drugs taken and thus reduce the risk of subsequent hospitalisation. To this end, this manuscript was written with the aim of identifying three advanced markers of hepatotoxicity that should be investigated in future in patients abusing various types of NPSs, or at least initially in preclinical studies, so that some concrete conclusions can be drawn.

## Figures and Tables

**Figure 1 ijms-24-09413-f001:**
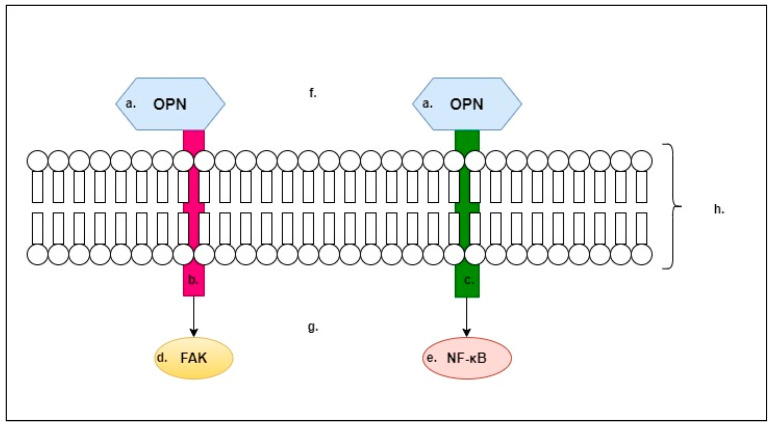
Mechanism of action of OPN. a—OPN; b—integrin; c—CD44 antigen; d—focal adhesion kinase (FAK); e—nuclear factor-kappa B (NF-κB); f—extracellular matrix; g—intracellular matrix; h—cell membrane. This figure describes the mechanisms by which OPN binds to two molecules, integrin (b) and CD44 (c). OPN (a) binding to integrin (b) causes activation of FAK (d), resulting in cell migration. This process also results in the activation of the FAK/Rac1/cell division control protein 42 (Cdc42)-GTPase pathway, which facilitates migration. OPN (a) after binding to CD44 (c) activates the NF-κB pathway (e). Activation of the NF-κB pathway is preceded by activation of the phosphatidylinositol 3 kinase (PI3K)/AKT pathway, activation of the mitogen-activated protein kinase (MAPK) signalling pathway and phosphorylation of the NF-κB inhibitor (IκB). These processes result in the activation of the NF-κB pathway.

**Figure 2 ijms-24-09413-f002:**
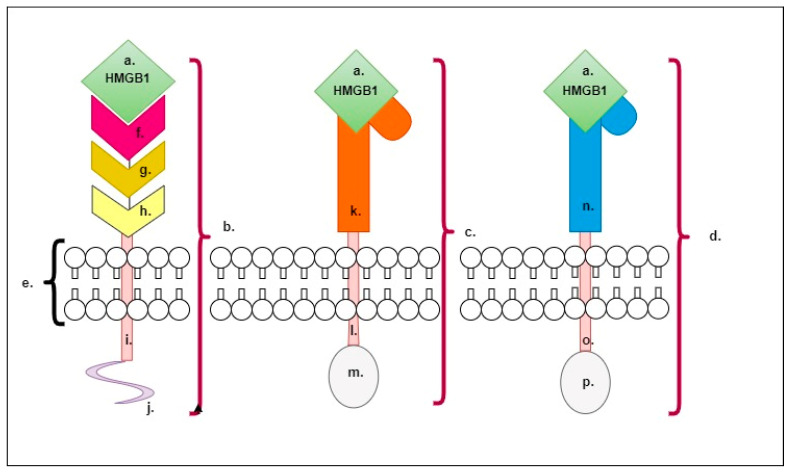
Mechanism of action of HMGB1. a—HMGB1; b—receptor for advanced glycation end product (RAGE); c—toll-like receptor 2 (TLR2); d—toll-like receptor 4 (TLR4); e—cell membrane; f, g, h—extracellular domains of RAGE; i—transmembrane-spanning domain; j—intracellular domain of RAGE; k, n—extracellular domains of TLR2/TLR4; l, o—single helix transmembrane domain; m, p—intracellular domains of TLR2/TLR4. This figure shows the 3 receptors for which HMGB1 is the ligand; these are RAGE (b), TLR2 (c) and TLR4 (d). RAGE is a type-I transmembrane protein that consists of three extracellular immunoglobulin-like domains (f–h), a single trans-membrane helix (i) and a short intracellular C-terminal domain (j). HMGB1 binding to the extracellular domain activates pro-inflammatory or anti-inflammatory pathways; this is dependent on the molecules involved in the process. TLR2 (c) and TLR4 (d) are composed of 3 domains—they contain N-terminal extracellular domains (k, n), transmembrane domains that are single helixes (l, o) and intracellular C-terminal domains (m, p). The fusion of HMGB1 with TLR2 and TLR4 results in the activation of the full pro-inflammatory response, the NLRP3 inflammasome and the NF-κB pathway and the synthesis of inflammatory cytokines.

**Figure 3 ijms-24-09413-f003:**
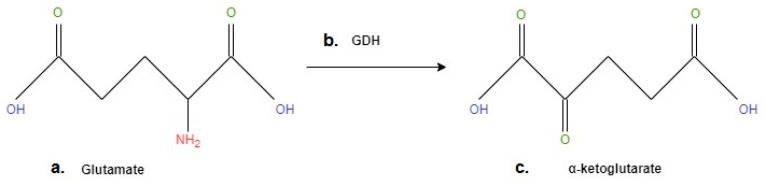
Mechanism of action of GDH. a—glutamate; b—GDH; c—α-ketoglutarate. This figure shows a diagram of the action of GDH (b). It is a process that involves a deamination mechanism. GDH (b) is the enzyme involved in the conversion of glutamate (a) to α-ketoglutarate (c). The second product of this reaction is ammonia.

**Table 1 ijms-24-09413-t001:** The most important research on GDH, OPN and HMGB1 in mental illness.

Mental Illness	Biomarker	Participants in Study	Results	Reference
Schizophrenia	OPN	Study group: 22 patients with schizophrenia Control group: -	Schizophrenic patients under long-term antipsychotics treatment had lower OPN levels than patients under short-term treatment (*p* = 0.021).	[35]
Schizophrenia	HMGB1	Study group: - 30 schizophrenic patients in the acute exacerbation phase - 29 schizophrenic patients in the remission phase15 healthy controls	HMGB1 levels were higher in study group than in healthy controls, independent of the phase of the disease (acute exacerbation vs. control—*p* = 0.05; remission vs. control—*p* = 0.002).	[101]
Schizophrenia	HMGB1	Study group: 115 patients with schizophrenia and 43 healthy controls	HMGB1 levels were higher compared with healthy controls (*p* < 0.001).	[102]
Schizophrenia	GDH	Study group: brain tissues taken at autopsy of 8 schizophrenic patients Control group: brain tissues taken at autopsy of 9 healthy people	Significant differences in GDH levels were observed in the prefrontal cortex (area 10) in schizophrenic patients compared with healthy controls (*p* < 0.01).	[208]
Major depressive Disorder	OPN	Study group: 1172 patients with MDD and 426 healthy controls	OPN levels in patients with MDD were significantly lower compared with healthy controls (*p* = 0.012).	[38]
Major depressive Disorder	OPN	Study group: 28 patients with MDD and 16 healthy controls.	OPN levels in patients with MDD were significantly lower compared with healthy controls. Ketamine significantly increased OPN levels on the 1st day (*p* < 0.001) and on the 3rd day (*p* < 0.001) after administration.	[40]
Major depressive disorder	GDH	Study group: 78 elderly patients with MDD (including 42 patients before treatment and 36 patients after treatment) and 29 healthy controls	Patients diagnosed with MDD had significant decrease in the activity of GDH compared with healthy controls (*p* < 0.0008).	[222]
Bipolar disorder (BD)	HMGB1	Study group: 17 patients diagnosed with BD and 16 healthy controls	HMGB1 levels were significantly higher in study group than in healthy controls (*p* < 0.0001).	[145]
Eating disorder	OPN	Study group: 20 patients diagnosed with anorexia nervosa and 78 healthy controls	OPN levels were significantly higher in patients with anorexia compared with healthy controls (*p* = 0.009).	[48]
Eating disorder	HMGB1	Study group: 11 patients diagnosed with anorexia nervosa (during the observation period and during the refeeding-resistant period) and 11 healthy controls	The average HMGB1 levels in healthy controls and participants during the observation period were significantly lower compared with group during the refeeding-resistant period (*p* < 0.005).	[148]
Autism	OPN	Study group: 42 autistic children Control group: 42 healthy children	The autistic children had significantly higher OPN levels compared with healthy controls (*p* = 0.02).	[49]
Autism	HMGB1	Study group: 42 autistic children Control group: 38 healthy children	The autistic children had significantly higher HMGB1 levels compared with healthy controls (*p* = 0.039).	[154]
Autism	HMGB1	Study group: 22 adult patients with autistic disorders and 28 healthy controls	The autistic patients had significantly higher HMGB1 levels than the healthy controls (*p* < 0.001).	[157]
Autism	GDH	Study group: 20 patients with autistic disorders and 20 healthy controls	The GDH activity was lower compared with healthy controls (*p* = 0.001).	[225]
Alcohol dependence	OPN	The retrospective group: 109 participants The prospective group: 95 participants (Included 44 patients with significant liver fibrosis and 65 patients with mild fibrosis.)	The patients with significant liver fibrosis had higher OPN levels compared with the patients with mild fibrosis (*p* < 0.001).	[53]
Alcohol dependence	HMGB1	Study group: 80 patients with alcohol use disorder and alcohol-associated liver disease	The patients with alcohol use disorder and alcohol-associated liver disease had higher HMGB1 levels compared with patients with alcohol use disorder without alcohol-associated liver disease (*p* = 0.0002).	[168]
Alcohol dependence	GDH	Study group: 238 alcoholic patients (and 141 healthy controls)	GDH levels were significantly higher in alcoholic patients compared with healthy controls (*p* < 0.0005).	[232]

## Data Availability

Not applicable.

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
