# Peer review of "Advanced Biomarkers of Hepatotoxicity in Psychiatry: A Narrative Review and Recommendations for New Psychoactive Substances"

_ijms, 2023, doi:10.3390/ijms24119413_

Round 1

Reviewer 1 Report

Golub et al extensively sumarized and discussed some advanced biomarkers of hepatotoxicity in psychiatry that should be borne in mind when conducting future research into the effects of new psychoactive substances on liver function. Overally, this manuscript is well written and deeply reviewed all manners regards the topic. However, their discussion, further investigation and current limitation are still needed to be extensive revised. Also, it would be better to have a table to summary all these biomarkes. There are 3 figures, their mechainism and describing are still limitted. 

Author Response

Dear Reviewer nr 1, 

Thank you very much for sending your comments related to our manuscript, as well as the associated positive word.

Comment 1: Golub et al extensively sumarized and discussed some advanced biomarkers of hepatotoxicity in psychiatry that should be borne in mind when conducting future research into the effects of new psychoactive substances on liver function. Overally, this manuscript is well written and deeply reviewed all manners regards the topic. However, their discussion, further investigation and current limitation are still needed to be extensive revised. Also, it would be better to have a table to summary all these biomarkes. There are 3 figures, their mechanism and describing are still limitted.

Reply 1: Following the recommended guidelines, in this manuscript the discussion has been extended to be able to point even more broadly to the validity of conducting this type of study in future in patients abusing different types of NPS.

Other changes that were made to the manuscript include the creation of a summary table, i.e. including the most important studies on individual markers of hepatotoxicity. The description of the figures as also pointed out by another reviewer has been expanded, including their legend.

Thank you again for giving us valuable advice for the review article we wrote.

Reviewer 2 Report

Authors reviewed about biomarkers of hepatotoxicity in psychiatry, especially focused on OPN, HMGB1 and GDH. This review was well-summarized and interesting. In Figures, legends to understand the schema are needed. Discussion about markers to predict disease severity might be helpful for clinicians. 

English was understandable. Minor grammatical check should be done.

Author Response

Dear Reviewer nr 2,

Thank you for giving us valuable advice for the review article we wrote.

Comment 1: Authors reviewed about biomarkers of hepatotoxicity in psychiatry, especially focused on OPN, HMGB1 and GDH. This review was well-summarized and interesting. In Figures, legends to understand the schema are needed. Discussion about markers to predict disease severity might be helpful for clinicians.

Reply 1: As you wrote about and as another reviewer pointed out - the discussion was expanded to emphasise the importance of the published article.

Reviewer 3 Report

Review Report for the Manuscript “Advanced biomarkers of hepatotoxicity in psychiatry: a narrative review and recommendations for new psychoactive substances

Rating the Manuscript

English Level: Is the English language appropriate and understandable?

Yes, English language in the manuscript is appropriate and understandable. 

General comments

In most parts of the manuscript the authors cite previously reported results, but they don’t explain the reasoning for those results. I think it’s better to reduce the number of citations that just show the results and focus on few examples and explain those in detail. 

Figures and figure captions need improved.

Overall Recommendation: 

Accept after Minor Revisions

Given below are the comments for each section of the manuscript.

Abstract

Define the term HMGB1 protein.

1.Introduction

1.     The liver parameters studied so far in a larger group of patients, which are simple liver enzymes, are nonspecific biomarkers, as they are found in many cell types and their increase is recorded with damage to almost every organ.

Briefly explain about simple liver enzymes.

2.Osteopontin (OPN)

1.     “In contrast, the interaction of OPN with the CD44 antigen results in the activation of the NF κB pathway, which, at the molecular level, initiates the expression of pro-inflammatory genes.”

Briefly discuss about NF kB pathway.

What pro-inflammatory genes are expression are initiated?

2.     “A study by Morales-Ibanez et al. found that OPN-deficient mice were protected from alcohol-induced liver damage and showed reduced expression of inflammatory cytokines.”

Which cytokines showed reduced expression.

3.     Moreover, as reported by Castello et al, OPN appears to be a key determinant of the interaction between tumour cells and the host microenvironment.

Briefly explain the role of OPN in the interaction between tumour cells and the host microenvironment.

2.1. OPN in psychiatry

1.     Higher levels of BDKRB1 and SPP1 (the gene encoding OPN) gene expression were observed in patients after a first psychotic episode (FEP) compared to healthy controls, and those taking risperidone had higher expression of these genes than all other patients studied.”

Define the terms BDKRB1 and SPP1.

2.     “Zhang et al. conducted a preclinicalstudy with which they demonstrated that bone inflammatory markers (including OPN) may play a role in the antidepressant effect of (R)-ketamine, as mice with major depressive disorder (MDD) showed a significant increase in OPN (and OPN/RANKL) in response to (R,S)-ketamine on days 1 and 3 after a single infusion of [38].”

This sentence is incomplete.

3.     “He also investigated other proteins known to interact with CD44, such as OPN.”

What other proteins interact with CD44?

2.2. OPN and use of psychoactive substances

1.     “One of the main complications of alcoholism is alcoholic liver disease. The review, which was conducted in 2021, presents the state of knowledge on the contribution of OPN to liver steatosis in the context of alcoholic liver disease and non-alcoholic steatohepatitis. This review reports that serum and liver OPN levels are elevated in patients with alcoholic liver disease. The same results are provided by studies conducted on animal models. In the course of alcoholic liver disease, increasing levels of OPN inhibit disease progression and iron deposition in the liver.”

What’s the reference for this paragraph? Is it reference 18? If so, include it in this paragraph as well.

2.     By the same mechanism, it also affects macrophage activation and on osteogenic and osteoclastic signalling in the medial palatal suture.

Here authors mention by the same mechanism, but they didn’t explain the mechanism earlier.

3.High-mobility group box 1 protein (HMGB1, HMG-1, amphoterin)

1.     “Under the influence of the interaction of HMGB1 with TLR2 or TLR4, there is activation of the full pro inflammatory response, the NLRP inflammasome3, activation of NF-κb and synthesis of inflammatory cytokines.”

Did you want to say “NLRP 3 inflammasome”?

2.     “In Fig. 2 we also present the structural structure of TRL2 and TRL4, which are classified as integral type I transmembrane proteins and are composed of three domains - an N-terminal domain located outside the membrane, a central transmembrane domain (single helix) crossing the membrane and a C-terminal domain located towards the cytoplasm.”

What do you mean by “structural structure of TRL2 and TRL4”?

3.1. HMGB1 in psychiatry

1.     “Thereview by Zhang et al. included 69 articles, of which a closer focus was on seven articles that met the criteria set by the authors.”

There should be a space between “the” and “review”.

3.2. HMGB1 and use of psychoactive substances

1.     “Various isoforms of HMGB1, such as disulfide-linked hyperacetylated HMGB1, were observed in the serum of both ethanol-treated mice and patients with alcoholic liver disease.”

Are the isoforms of HMGB1 present only in this case? What about other studies where they studied the effect of alcohol? 

2.     “Orio et al. showed that there are gender-related differences in the peripheral inflammatory response to alcohol in young people, with increased levels of alcohol risk-related molecules such as HMGB1 observed in female drinkers.”

 Do you know the reason for the gender related differences? 

3.     “Almost all the studies referred to in the 2015 review by Gangemi et al. showed that HMGB1 levels are elevated in smokers and COPD patients.”

Define the term COPD.

4.     Glutamate dehydrogenase (GDH, GLDH)

1.     “In a study by Michalak et al, lung cancer chemotherapy induced an increase in glutamate in peripheral blood mononuclear cells (PBMCs), and its serum concentration increased GDH activity in PBMCs.”

Have they discussed about the mechanism for this reaction?

2.     “Furthermore, there are reports that the mitochondrial enzyme GDH1 is commonly elevated in human cancers.”

Is it elevated in all types of cancers?

Figures

Quality of the figures could be improved.

Figure 1: 

Figure caption should be more informative.

Also, authors need to label the figure properly. For example, what the squares and circles represent?

Figure 2: Figure caption should be more informative and the labeling needs to be improved.

References:

Some of the references are more than 10 years old. It they don’t contain important information authors could replace these with new references. 

References: 

12,20,25,26,28,29,31,35,45,48,52,58,59,60,61,65,67,72,73,78,80,83,85,86,89,92,113,147,156,157,164,187,201,202,206,207,210,211,215,217,218,227,231,232,233,234,235,236,242,

Author Response

Dear Reviewer nr 3,

Thank you for your positive words regarding the written manuscript.

Comment 1: Define the term HMGB1 protein.

Reply 1: All abbreviations in the manuscript have been expanded, while a list of abbreviations is provided at the end of the manuscript.

Comment 2: Briefly explain about simple liver enzymes.

Reply 2: It is indicated in the text what simple parameters are referred to.

Comment 3: Define the terms BDKRB1 and SPP1.

Reply 3: Abbreviation extensions have been added.

Comment 4: There should be a space between “the” and “review”.

Reply 4: A space has been added where indicated.

Comment 5: Define the term COPD.

Reply 5: Abbreviation extension has been added.

Comment 6: Quality of the figures could be improved.

Reply 6: The quality of figures has been improved.

Comment 7: Figure 1: Figure caption should be more informative.

Also, authors need to label the figure properly. For example, what the squares and circles represent? Figure 2: Figure caption should be more informative and the labeling needs to be improved.

Reply 7: The description of the figures as also pointed out by another reviewer has been expanded, including their legend.

Comment 8: Did you want to say “NLRP 3 inflammasome”?

Reply 8: The indicated abbreviation has been corrected.

Comment 9: What do you mean by “structural structure of TRL2 and TRL4”?

Reply 9: The indicated abbreviation have been extended.

Comment 10: What’s the reference for this paragraph? Is it reference 18? If so, include it in this paragraph as well.

Reply 10: The reference has been added.

Comment 11: What other proteins interact with CD44?

Reply 11: We have indicated what kind of proteins they are.

Comment 12: Briefly discuss about NF kB pathway. What pro-inflammatory genes are expression are initiated?

Reply 12: Activation of the NF kB pathway is described under the first figure. It is indicated which genes these are.

Comment 13: Which cytokines showed reduced expression?

Reply 13: We have indicated what these cytokines are.

Comment 14: Briefly explain the role of OPN in the interaction between tumour cells and the host microenvironment.

Reply 14: The description has been extended by the indicated aspect.

Comment 15: This sentence is incomplete.

Reply 15: The indicated sentence was edited by shortening it.

Comment 16: Here authors mention by the same mechanism, but they didn’t explain the mechanism earlier.

Reply 16: The sentence has been corrected so that it is clear what mechanism is involved.

Comment 17: Are the isoforms of HMGB1 present only in this case? What about other studies where they studied the effect of alcohol?

Reply 17: The passage quoted by the reviewer refers to a 2018 review paper on HMGB1 and liver disease. No other papers were found that examined HMGB1 isoforms in the context of ethanol consumption.

Comment 18: Do you know the reason for the gender related differences?

Reply 18: Gender differences were pointed out.

Comment 19: Do you know the reason for the gender related differences?

Reply 19: The mechanism has been explained.

Comment 20: Is it elevated in all types of cancers?

Reply 20: Specific types of cancer are indicated.

Comment 21: Some of the references are more than 10 years old. It they don’t contain important information authors could replace these with new references.

Reply 21: The indicated items were reviewed and replaced with newer ones where it was possible and sensible. In other words, articles were replaced with newer ones where there was a general mention of hepatotoxicity markers.

Reviewer 4 Report

Reviewer

 Initial comments

 Review

Advanced biomarkers of hepatotoxicity in psychiatry: a narrative review and recommendations for new psychoactive substances

 Aniela Goluba, Michal Ordaka, Tadeusz Nasierowskib and Magdalena Bujalska-Zadroznya

 Author Contributions: Conceptualization, M.O.; writing—original draft preparation, A.G. and

M.O.; writing—review and editing, A.G. and M.O.; supervision M.O., T.N., and M.B.Z. All authors

have read and agreed to the published version of the manuscript.

 This paper is very important, because the use of drugs in psychiatry is increasing and the toxic effects of these drugs need to be carefully verified, mainly the effects on the liver and in patients diagnosed with hepatitis, mainly type C hepatitis, which is currently being treated with drugs with good results. Thus, this paper brings a great contribution to all health professionals who use these drugs, and also to psychiatrists.

References

 1. Peacock, A.; Bruno, R.; Gisev, N.; Degenhardt, L.; Hall, W.; Sedefov, R.; White, J.; Thomas, K.V.; Farrell,

M.; Griffiths, P. New psychoactive substances: challenges for drug surveillance, control, and public health

responses. Lancet 2019, 394, 1668-1684, doi:10.1016/S0140-6736(19)32231-7.

 248. Anderson, C.; Morrell, C.; Marchevsky, D. A novel psychoactive substance poses a new challenge in the

management of paranoid schizophrenia. BMJ Case Rep 2015, 2015, doi:10.1136/bcr-2015-209573

 There are 248 references, so, this paper also draws attention due to the number of references, extensive, with a very restricted number of authors, different from the way we have reviewed review articles.

 5. Recommendations on hepatotoxicity testing of new psychoactive substances

 This item 5 is very pertinent, because in addition to the important recommendations, it brings a summary at the end, so that we can better fix this extensive review article.

 Abbreviations – The paper is very extensive and has many acronyms, and a list of abbreviations would be advisable for a better understanding of the text.

 Title

Advanced biomarkers of hepatotoxicity in psychiatry: a narrative review and recommendations for new psychoactive substances

 Comment: It is suitable

Abstract:

 Comment:

 HMGB1 protein….

 Please, also write in full here in the summary as already mentioned in item 3.

 3. High-mobility group box 1 protein (HMGB1…HCV infection,.....

 Please, put HCV in full.

1. Introduction  

Comment:

 HCV infection...

 Please, put HCV in full.

 2. Osteopontin (OPN)

Comment: It is suitable

 2.1. OPN in psychiatry

 Comment: It is suitable

 2.2. OPN and use of psychoactive substances

 Comment: It is suitable

3. High-mobility group box 1 protein (HMGB1, HMG-1, amphoterin)

Comment: It is suitable

3.1. HMGB1 in psychiatry

Comment: It is suitable

3.2. HMGB1 and use of psychoactive substances

Comment: It is suitable

4. Glutamate dehydrogenase (GDH, GLDH)

Comment: It is suitable

4.1. GDH in psychiatry

Comment: It is suitable

4.2. GDH and use of psychoactive substances

Comment: It is suitable

5. Recommendations on hepatotoxicity testing of new psychoactive substances

Comment: It is suitable

References

Comment: It is suitable

Thank you

Author Response

Dear Reviewer nr 4,

Thank you for your positive words regarding the written manuscript.

Comment 1: This paper is very important, because the use of drugs in psychiatry is increasing and the toxic effects of these drugs need to be carefully verified, mainly the effects on the liver and in patients diagnosed with hepatitis, mainly type C hepatitis, which is currently being treated with drugs with good results. Thus, this paper brings a great contribution to all health professionals who use these drugs, and also to psychiatrists.

There are 248 references, so, this paper also draws attention due to the number of references, extensive, with a very restricted number of authors, different from the way we have reviewed review articles.

Reply 1: The extensive number of articles is due to the desire to show the subject under study in the best possible context. This number of authors is due to the fact that the work carried out took more time compared to the manuscripts written previously.

Comment 2: Abbreviations – The paper is very extensive and has many acronyms, and a list of abbreviations would be advisable for a better understanding of the text.

Reply 2: In line with the advice received, the names of first-time abbreviations have been expanded in the text, and a list of abbreviations has been created at the end of the manuscript.

Comment 3: This item 5 is very pertinent, because in addition to the important recommendations, it brings a summary at the end, so that we can better fix this extensive review article.

Reply 3: Following the advice received, the discussion has been expanded.

Comment 4: Please, also write in full here in the summary as already mentioned in item 3.

Please, put HCV in full (abstract, introduction).

Reply 4: Following the advice received, this abbreviation has been expanded.

Reviewer 5 Report

Some minor issues should be clarified:

The abstract should explain how the authors chose HCV as the study virus. It is not clear.

Abbreviations should be specified in tables and figures.

Figure 2 is not explained by itself, the authors should explain the different elements of the graph.

It would be beneficial to generate a table and a graph with the main observations detailed during the manuscript as a summary.

It would be advisable to reduce the different sections by establishing direct objectives that allow a fluent reading of the manuscript.

Minor editing of English language required

Author Response

Dear Reviewer nr 5,

Thank you for your positive words regarding the written manuscript.

Comment 1: The abstract should explain how the authors chose HCV as the study virus. It is not clear.

Reply 1: A sentence on HCV infection was added to the abstract.

Comment 2: Abbreviations should be specified in tables and figures.

Reply 2: As requested by the other reviewers, a list of abbreviations has been included at the end of the manuscript.

Comment 3: Figure 2 is not explained by itself, the authors should explain the different elements of the graph.

Reply 3: As the other reviewers also asked, individual figures were described in more depth.

Comment 4: It would be beneficial to generate a table and a graph with the main observations detailed during the manuscript as a summary.

Reply 4: Following an additional suggestion from yet another reviewer, a summary table was created, i.e. including the most important articles.

Comment 5: It would be advisable to reduce the different sections by establishing direct objectives that allow a fluent reading of the manuscript.

Reply 5: The fluent reading of the manuscript was improved by doing several things, i.e. done, among other things, on the basis of the other 4 reviews. In line with additional suggestions from the other reviewers, references were changed to newer ones in many places. In order to increase the clarity of the manuscript, an additional description was placed under each figure. In addition, to illustrate the subject matter of the manuscript, a summary table was created that includes the most important articles indicated in the manuscript, so that the interested party can know what the research results published in them are about. The quality of the figures has been improved. In order to provide an even more in-depth presentation of the significance of the topic obtained, the discussion of the review carried out has been expanded with limitations, as suggested by the other reviewers. All abbreviations have been listed and expanded at the bottom.

Round 2

Reviewer 1 Report

Thanks for accepting and responding all my concerns. There is no further comments. The work can be acceptable now!